# Transoral Robotic Surgery for Pharyngeal and Laryngeal Cancers—A Prospective Medium-Term Study

**DOI:** 10.3390/jcm10050967

**Published:** 2021-03-02

**Authors:** Chen-Chi Wang, Wen-Jiun Lin, Yi-Chun Liu, Chien-Chih Chen, Shang-Heng Wu, Shih-An Liu, Kai-Li Liang

**Affiliations:** 1School of Medicine, National Yang Ming Chiao Tung University, Taipei 112, Taiwan; b9202080@gmail.com (Y.-C.L.); saliu@vghtc.gov.tw (S.-A.L.); kellyliang1107@gmail.com (K.-L.L.); 2School of Speech Language Pathology & Audiology, Chung Shan Medical University, Taichung 40121, Taiwan; 3Department of Audiology and Speech-Language Pathology, Asia University, Taichung 41354, Taiwan; 4Department of Otolaryngology-Head & Neck Surgery, Taichung Veterans General Hospital, Taichung 40705, Taiwan; kgjack2001@gmail.com (W.-J.L.); entwsh@gmail.com (S.-H.W.); 5Department of Radiation Oncology, Taichung Veterans General Hospital, Taichung 40705, Taiwan; chiencheh@gmail.com; 6Ph. D. Program in Translational Medicine, National Chung-Hsing University, Taichung 40227, Taiwan

**Keywords:** cancer, chemotherapy, larynx, pharynx, radiotherapy, robot, survival, transoral robotic surgery, neoadjuvant chemotherapy, adjuvant radiotherapy

## Abstract

Transoral robotic surgery (TORS) has been used for treating pharyngeal and laryngeal cancers for many years. However, the application of neoadjuvant chemotherapy (NACT) before TORS, the sparing rate of adjuvant irradiation after TORS, and the long-term oncologic/functional outcomes of TORS are seldom reported. From September 2014 to May 2018, 30 patients with clinical T1 to T3 cancers of oropharynx (7), larynx (3), and hypopharynx (20) were prospectively recruited for TORS in a tertiary referral medical center. Twelve (40%) patients had clinical early stage (I or II) disease, and 18 (60%) patients had late-stage (III or IV) disease. All 30 patients were suggested to receive TORS with neck dissection. Cisplatin-based NACT was given to 11 patients before the surgery, and it led to a 100% reduction in tumor size. Only 40% of patients needed adjuvant irradiation with a mean dosage of 5933 cGY after TORS. After a mean follow up of 38.9± 14.7 months, the Kaplan–Meier method estimated 5-year disease-specific survival, and organ preservation was 86.3% and 96.2%, respectively. Twenty-five patients were alive without tracheostomy and tube feeding. We found that NACT is a potential method for facilitating tumor resection and TORS effectively de-escalated adjuvant irradiation with a satisfactory 5-year survival and functional outcomes.

## 1. Introduction

Organ preservation is a key objective in the nonsurgical management of pharynx and larynx cancers. Since the Veteran’s Affairs (VA) larynx preservation study [1], radiation combined with chemotherapy has become a popular organ preservation management technique for pharynx and larynx cancers. However, organ preservation does not always result in functional preservation. Acute toxicity of chemoradiation such as mucositis had a high rate of 80% in the results from 6181 patients in a systematic literature review by Trotti A et al. [2]. Unfortunately, acute toxicities may persist chronically, although it is often under-reported or underestimated according to a trial report [3]. It has been estimated that 46% to 63% of head and neck cancer survivors demonstrate some degree of late dysphagia after high-intensity radiotherapy with or without chemotherapy [4,5,6]. Late dysphagia always has a significant impact on quality of life [7].

The aforementioned acute and chronic complications of non-surgical treatment experienced by patients have spurred the exploration of minimally invasive surgeries to avoid negative functional consequences. Transoral laser microsurgery (TLM) has thus become a popular approach for laryngeal and pharyngeal cancers [8,9,10]. However, the convenience of surgery is limited by the line of sight of microscopic view via a narrow laryngoscope. Tumor is usually removed by piecemeal resection, and resection margin evaluation becomes difficult. In addition, the hemostatic function of the CO2 laser beam is less effective than electrocautery. Since the development of the da Vinci robot, transoral robotic surgery (TORS) has been adopted for treating pharyngeal and laryngeal cancers, because TORS may offer the benefits of overcoming the aforementioned limitations in CO2 laser surgery [11,12,13,14]. In January 2010, the U.S. Food and Drug Administration approved TORS for use in benign and malignant diseases of the pharynx and larynx [15]. Although robotic technologies continue to evolve and applications have been expanded, many questions remain unanswered, such as the feasibility of neoadjuvant chemotherapy (NACT) before TORS and how much the de-intensification of radiotherapy could be achieved after TORS [16,17]. Medium-term and long-term survival data have seldom been reported. This prospective study presents the medium-term survival results of TORS for pharyngeal and laryngeal cancers and the effects of applying NACT and de-intensification of radiotherapy for preserving satisfactory laryngopharyngeal function after treatment.

## 2. Materials and Methods

From September 2014 to May 2018, according to the Cancer Staging System 7th Edition of American Joint Committee on Cancer (AJCC), 30 patients who were diagnosed with T1 to T3 upper aerodigestive tract (UADT) cancers involving oropharynx, larynx or hypopharynx were prospectively enrolled in a cancer study conducted by a tertiary referral medical center and sponsored by Taiwan’s Ministry of Health and Welfare (program number: MOHW106-TDU-B-211-144002). Twelve (40%) patients had clinical early stage (I or II) disease, and 18 (60%) patients had late-stage (III or IV) disease. The expense of da Vinci surgery was covered by the program, and the limited budget was sufficient to enroll 30 patients into the study. The study was approved by the Institutional Review Board of Taichung Veterans General Hospital (IRB TCVGH No: SF13226) and met the guidelines of our responsible governmental agency and the precepts established by the Helsinki Declaration. Thirty patients were recruited to receive TORS with or without NACT and neck dissection. The demographic data, peri-operation information, adjuvant therapy, and postoperative follow-up data were recorded in our database, and medium-term treatment outcomes were analyzed to elucidate the effectiveness of this surgical organ preservation strategy.

### 2.1. TORS and Neck Dissection with or without NACT

All patients diagnosed with T1 to T3 UADT cancers received TORS-assisted surgery using the da Vinci robot Si system (Intuitive Surgical, Sunnyvale, CA, USA) after providing informed consent. For some patients with T3 stage tumors and patients with early T1-T2 tumors who could not receive TORS promptly after diagnosis, Cisplatin-based NACT was given before TORS. All TORS were performed by the first author with assistance from the second author, under general anesthesia without tracheostomy, as described in detail elsewhere [18,19]. Patients’ mouths were opened with the Laryngeal Advanced Retractor System (LARS; Fentex, Tuttlingen, Germany) to achieve adequate exposure of the surgical sites. An example of TORS for hypopharyngeal cancer is shown in Figure 1 and Appendix A. None of the patients finished TORS until all frozen section margins were negative for malignancy during the surgery. The cut margins of permanent specimens were dichotomized to “margin free” and “margin with concern” (close margin, margin that is difficult to evaluate, or only microscopically identified cancer cells on the margin). All patients were recommended to receive simultaneous neck dissection ipsilateral to the cancer side, at minimum, following TORS. However, if clinical N0 patients refused neck dissection, then watchful observation was employed. If there was clinical evidence of bilateral neck metastasis, bilateral neck dissections were performed. 

### 2.2. Adjuvant Therapy and Active Surveillance after TORS

After surgery, adjuvant therapy was decided according to the pathologic results, National Comprehensive Cancer Network (NCCN) practice guidelines in Oncology, consensus achieved in a multidisciplinary combined meeting for each case, the patients’ preferences, and the final decision. In our hospital, for patients with T3 tumors, N2-N3 nodal disease, perineural invasion, lymphovascular invasion, extracapsular extension, or positive margins, postoperative adjuvant radiotherapy ± chemotherapy is recommended. After treatment, the patients were followed up regularly by physical examination, including a flexible laryngoscope every month in the first year, every 2 months in the second year, every 3 months in the third year, and every 4 to 6 months thereafter. Image studies, such as chest X-ray, computed tomography (CT)/magnetic resonance imaging (MRI), abdominal echography, whole body bone scan, and Positron Emission Tomography (PET/CT), were used to survey local recurrence, cervical regional recurrence, or distant organ metastasis every 6 to 12 months according to the physical examination results. After medium-term follow-up, survival analysis by the Kaplan–Meier method was used to determine the overall, disease-specific, and recurrence-free survival. Factors that may be associated with survival, such as age, gender, tumor sites, clinical stages, pathologic stages, cut margins, neoadjuvant chemotherapy (NACT), adjuvant radiotherapy, local recurrence, regional recurrence, and distant metastasis, were described and analyzed by a t-test, Chi-squared test, and Fisher’s exact test. The survival curves were compared for different tumor sites, NACT, and post-TORS adjuvant radiotherapy by a log-rank test. In addition, for surviving patients, medium-term functional evaluations, such as larynx preservation rate, tracheostomy dependence rate, tube feeding rate, and phonation function, recorded by Voice Handicap Index-10 (VHI-10) [20] and Functional Outcome Swallowing Scale (FOSS) [21] were also recorded.

### 2.3. Purpose of the Study

The main purpose of the study was to establish the medium-term survival outcome of TORS surgery and the rate of de-escalating adjuvant radiotherapy. In addition, we investigated the applications of NACT before TORS and the functional results of this minimally invasive organ-preserving surgery. 

## 3. Results

Successful TORS with/without neck dissection was achieved safely in all 30 patients without tracheostomy. Their ages ranged from 38 to 74 with a mean ± SD of 55.9 ± 10.1. Twenty-eight patients were male, and two patients were female. Seven (23.33%) patients had oropharyngeal cancers, 20 (66.67%) patients had hypopharyngeal cancers, and three (10.00%) patients had laryngeal cancers. Thirteen (43.33%) patients had clinical T1 tumor, 11 (36.67%) patents had clinical T2 tumor, and six (20%) patients had clinical T3 tumor. Overall, 12 (40%) patients had clinical early stage (I or II) disease, and 18 (60%) patients had clinical late-stage (III or IV) disease. The peri-operative data are summarized in Table 1.

### 3.1. NACT and TORS

Before TORS, 11 (36.67%) patients received cisplatin-based NACT. Ten patients received C-D-FL-ME (Cisplatin 60 mg/m^2^ on day 1, Docetaxel 50 mg/m^2^ on day 8, 5-FU 2500 mg/m^2^ + Leucovorin 250 mg/m^2^ on day 15, MTX 30 mg/m^2^ + Epirubicin 30 mg/m^2^ on day 22) and one patient received P-FL-ME (Cisplatin 60 mg/m^2^ on day 1, 5-FU 2500 mg/m^2^ + Leucovorin 250 mg/m^2^ on day 8, MTX 30 mg/m^2^ + Epirubicin 30 mg/m^2^ on day 15). The treatment duration of NACT ranged from 0.5 to 3 cycles with a mean of 2.2 cycles. The tumor in one of the patients who received C-D-FL-ME responded quickly and well after the injection of Docetaxel, and the patient received TORS after only half a course of NACT. All primary tumors responded to NACT. Six (54.5%) patients had complete remission proved by pathologic report of pT0 after surgery, and in the other five patients, the tumor was downsized to the pT1 stage. For those patients with a complete response after NACT, we still performed TORS, because we think if we could prove there is no tumor residual after NACT by minimal invasive surgery, the following radiotherapy to the primary site could be spared. This management would be an innovation which provides benefits to the patients.

Eight of the 11 patients had initial clinical N1 and N2 stages on computed tomography. After NACT, all suspected lymph nodes were smaller but not completely disappeared. However, four patients with suspected clinical N1 disease were revealed to be pathologically N0 after neck dissection. The response to NACT at T & N stages is summarized in Table 2. Although primary tumors responded to NACT, the excision extent was at least 1 cm from the original tumor border before NACT, according to the imaging studies and medical records. For example, the pre- and postoperation endoscopic images of a patient with T3 hypopharyngeal cancer who received NACT before TORS are shown in Figure 2. During the TORS, frozen sections were also performed on margins (superior, inferior, medial, lateral, and base) even in cases of complete remission to ensure that the margins were clear before completion of the surgery.

### 3.2. Nevk Dissection and Post-TORS Complications

Twenty-four patients received ipsilateral neck dissection, while three patients received bilateral neck dissection. Three patients with clinical N0 disease declined neck dissection due to various concerns. Twenty-four patients received level II, III, and IV functional or modified radical neck dissection. Three patients received further level V dissection in addition to level II to IV dissection. In total, 765 lymph nodes were harvested from 27 patients. The lymph nodes harvested from each neck dissection ranged from 7 to 49 with mean ± SD of 26.4 ± 10.0. In 27 patients receiving neck dissection, 10 (37%) patients had cancer metastasis in 1 to 10 lymph nodes. The number (mean ± SD) of positive lymph nodes in the 10 patients was 2.6 ± 2.9. Especially, 3 of the 10 patients with positive neck metastasis also revealed extranodal extension of the carcinoma.

After the surgery, there were no severe complications, such as airway emergency, life-threatening bleeding, or fistula formation. Only one patient with hypopharyngeal cancer had delayed wound oozing on the 11th day after surgery, and oozing stopped spontaneously after observation and intravenous tranexamic acid injection. Another patient had mild pneumonia that resolved after antibiotic treatment. 

### 3.3. Post-TORS Adjuvant Radiotherapy

After TORS ± neck dissection, pathologic staging was performed, and the results are summarized in Table 1. For the specimen of primary tumors, 23 (76.67%) patients had a free margin, and seven (23.33%) patients had a margin with concern (a close margin < 1 mm in two patients, a margin that is difficult to evaluate in two patients, or only microscopically identified cancer cells on the margin in three patients).Twelve (40%) patients received adjuvant radiation ± chemotherapy. (Table 1) Only one patient received the full dosage of 7000 cGy irradiation, and de-escalation of radiation was performed on the other 11 patients. The radiation dosage ranged from 5000 to 7000 cGY with a mean ± SD of 5933 ± 599 cGy. Three patients received salvage radiation ± chemotherapy after patients had loco-regional recurrence (radiation dose mean ± SD of 6733 ± 230). Overall, at the time of writing, irradiation could be reduced or avoided in 15 (50%) of the 30 patients during medium-term follow-up. 

### 3.4. Medium-Term Survival Data

The patients were followed up from 8 to 67 months with a mean± SD of 38.9± 14.7months. Five patients died during follow-up. One patient died of second primary esophageal cancer with mediastinal metastasis. One patient died of locoregional recurrence, and the other three patients died of distant metastasis with local or regional recurrence. The duration of survival of expired patients after surgery ranged from 8 to 51 months with a mean of 23.8± 16.9 months. 

Four patients had local recurrences at the primary operation sites (at 7, 16, 20, and 41 months after TORS, respectively). Two patients with hypopharyngeal cancers had very small T1 lesions that could be salvage resected by transoral laser microsurgery (16 and 41 months after TORS). One patient with hypopharyngeal cancer received total laryngectomy for local recurrence (20 months after TORS) before death, and he was the only patient in this cohort whose larynx could not be preserved. The last patient with laryngeal cancer received irradiation to preserve the organ (7 months after TORS), but died of distant metastasis. 

Five patients had neck regional recurrence. Three patients had contralateral neck recurrenceat postoperation at 1, 11, and 13 months, respectively. Two patients had ipsilateral neck recurrence post operation at 2 and 19 months, respectively. Four patients had simultaneous or subsequent distant metastasis (at 1, 13, 19, and 41 months after TORS respectively). A patient with retropharyngeal LN metastasis was successfully treated by second TORS with adjuvant chemoradiation. 

In the follow-up, three patients had cancers at other sites of the aerodigestive tract. The aforementioned patient with esophageal cancer and mediastinal metastasis died. A patient with hypopharyngeal ca had a second tonsil cancer that was successfully treated by a second TORS. A patient with primary tonsil ca developed tongue, hypopharynx, and contralateral tonsil ca that could be saved by surgery, including TORS without irradiation. 

The data of the 11 hypopharyngeal cancer patients who received NACT need to be elaborated, and they are summarized in Table 2. There was only one patient having recurrence at the primary site after 20 months follow up. The other 10 patients had no local recurrence. Although the number of patients in the specific group was quite small, the preliminary data indicate that our treatment strategy is a feasible alternative to conventional NACT with following radiotherapy. In the specific subgroup of six patients with complete primary tumor remission after NACT, a patient (no.10) still received 6000 cGy adjuvant radiotherapy because of the initial stage IV disease. Unfortunately, he still died from local and neck recurrence even after aggressive salvage total laryngectomy. Another patient (no.21) with initial stage III disease received 5000 cGy adjuvant radiotherapy. The other three patients (no. 17, 29, and 30) with initial stage III and one patient (no. 23) with initial stage II chose to be closely followed up. The aforementioned five patients (no. 17, 21, 23, 29, and 30) are alive without evidence of recurrence. 

The clinical factors and survival rates of the 30 patients are summarized in Table 1. In brief, only regional neck recurrence and distant metastasis were factors significantly (*p* ≤ 0.001) associated with overall survival. The other factors, namely, age, gender, tumor site, tumor stages, margin with concern, local recurrence, NACT, and radiation therapy, were not significantly associated with survival. The curves of overall, disease-specific, recurrence-free survival and organ preservation are shown in Figure 3. The estimated 5-year overall survival rate was 75.5%; the disease-specific survival rate was 86.3%; the recurrence-free survival rate was 71.1%; and organ preservation rate was 96.2%. The curves of disease-specific survival at different tumor sites are shown in Figure 4. The estimated disease-specific survival rate was 89.4% for hypopharyngeal cancer, 85.7% for oropharyngeal cancer, and 67% for laryngeal cancer. There was no significant difference among the three different tumor sites. (*p* = 0.493; log-rank test). For patients with/without NACT, and with/without radiotherapy, there was no significant difference in disease-specific survival with *p* = 0.557 and *p* =0.453, respectively (log-rank test).

### 3.5. Medium-Term Functional Data

In the cohort of 30 patients, preservation of the larynx was achieved in all but one patient. One patient required tracheostomy after laryngeal cancer recurrence before death. The tracheostomy-free rate was 93.3%. The laryngeal preservation rate was 96.7%, or 95.6% (22/23) if the seven patients with cancer of the oropharynx were excluded. Among the 25 alive patients during follow-up, none was dependent on tracheostomic or feeding tubes. The mean ± SD of VHI-10 (1.5 ± 4.0) and FOSS (0.1 ± 0.3) shown in Table 1 revealed that the treated patients had a good preservation of voice and swallowing function.

## 4. Discussion

### 4.1. Long-Term Results of TORS Were Rarely Reported

TORS ± adjuvant radiotherapy has been adopted in the treatment of cancers of the pharynx and larynx for longer than a decade. However, according to the Cochrane Database of Systematic Reviews, no randomized control trials have ever been conducted to compare TORS with (chemo)radiotherapy or to assess the effects of TORS with de-intensified adjuvant (chemo)radiotherapy [16,17]. In addition, Castellano A and Sharma A performed a systematic review and found that most studies in their analysis had short-term (1 to 2 years) follow-up after treatment [22], and the long-term data of TORS have seldom been reported. After preliminarily reporting the safety and feasibility of TORS in early T1-T2 upper aero-digestive tract cancers [18,19], our team started a 30-patient cohort prospective study in September 2014. The primary tumor size was extended to T3 cancer, and NACT was applied in some of our patients. Although the real follow up of each patient did not reach 5 years, we believe our medium-term follow-up data support the application of TORS in head and neck cancers.

### 4.2. TORS for Tumors in Different Sites of Pharynx and Larynx

The most well-established use of TORS has been in the treatment of oropharyngeal ca. While a number of studies have analyzed the use of TORS in oropharyngeal cancer, there has been relatively little discussion of TORS for hypopharyngeal cancer or laryngeal cancer. In this study, primary tumors from three sites could be safely resected by TORS with minimal complications. In the survival analysis, there was no significant difference in survival among the three different sites. The results are consistent with the results of our preliminary retrospective study [18,19] and suggest that hypopharyngeal cancer and laryngeal cancer could be treated with TORS as effectively as oropharyngeal cancer. 

### 4.3. Post-TORS Adjuvant Radiotherapy

According to a multi-institutional study on TORS, mainly for oropharyngeal cancers (88.8% of studied subjects), de Almeida JR et al. [23] reported that 52.6% of patients received adjuvant radiotherapy ± chemotherapy with a 2-year overall survival rate of 91% and a disease-specific survival rate of 94.5%. In our study, 40% of the patients received adjuvant radiotherapy ± chemotherapy, and the survival curve was comparable to the aforementioned results after 2 years of follow-up. The mean irradiation dose to pharyngeal constrictors, the floor of mouth, and other extra-laryngeal muscles was confirmed to be associated with an increased risk of dysphagia [24,25]. The low rate and de-intensification of adjuvant irradiation may explain the good functional results in our survived patients. The average dosage of irradiation in our 12 patients was about 6000 cGY, and thus this dosage could be a useful de-intensification reference value in future research. According to de Almeida JR et al. [23], tumors arising in the tonsils were associated with better survival than tumors from other sites (*p* = 0.01). However, there was no significant difference in our study. Our study, therefore, may inspire more studies on using TORS for hypopharyngeal and laryngeal cancers in addition to oropharyngeal cancer. In the aforementioned study of de Almeida JR et al. [23], tumor stages and cut margins were not associated with disease-specific survival. Our study confirmed these results. 

### 4.4. NACT and TORS

A secondary aim of the current study was to study the effectiveness of neoadjuvant chemotherapy (NACT) before TORS. NACT was used in the landmark Veteran’s Affairs (VA) larynx preservation study and shifted the paradigm of head and neck cancer treatment [1]. However, the role of NACT remained controversial in organ-preserving chemoradiation management. In a systematic review and meta-analysis conducted by Vidal et al. [26], the authors concluded that there was no beneficial effect of NACT plus chemoradiation on overall survival. Many studies used NACT as a form of chemoselection to identify patients who would likely respond to chemoradiation [27,28]. However, a meta-analysis conducted by Kiong KL [29] suggested that patients with poor NACT response will have a poorer response to chemoradiation, but a good NACT response does not guarantee a favorable outcome following chemoradiation. Therefore, for patients with a good response to NACT, minimally invasive surgery, such as TORS, is not an unreasonable option. Indeed, in most solid tumors, the value of NACT is to downsize the primary tumor to facilitate resection and organ preservation [30,31]. Based on this notion, NACT may be more valuable when used before minimally invasive surgery for a tumor located in a very small space, such as the pharynx and larynx. The downsized primary tumor can be more easily manipulated during the surgery. In a study by Park YM et al. [32], NACT was used before TORS, and they found that it helped to secure more than 70% of negative surgical margins. Our results support this finding, and patients had a higher (81.82% vs. 73.68%) rate of pathology-free cut margins in the NACT. However, there was no significant improvement in overall survival, disease-specific survival, recurrence/distant metastasis rate, irradiation rate, organ preservation, or functional preservation. However, from a different perspective, patients receiving NACT had a higher clinical T-stage without worse prognosis, which shows the potential benefit of NACT before TORS. However, NACT regimens vary among different institutes, and the application of our regimen in other places requires further investigation. Furthermore, possible related toxicities of NACT should be considered, although our patients could tolerate it and subsequently received TORS without issue.

### 4.5. TORS vs. Other Transoral Microsurgery

Minimally invasive surgery has evolved over the last two decades for treating head and neck cancers, and the most popular one is TLM, as mentioned in the introduction. According to the review of Suarez C and Rodrigo JP [33], TLM could be considered the method of choice for treating early T1-T2 laryngeal cancer, and could also be considered as an alternative to other organ preservation protocols in early stage I and II oroharyngeal and hypopharyngeal carcinoma. In their review, the laryngeal preservation rate ranged from 62% to 97% in laryngeal and hypopharyngeal cancer. Five-year survival rates were 60–75% in patients of stage I-II hypopharyngeal cancer and 35–45% in patients with stage III-IV hypopharyngeal cancer [34,35,36]. In the study of Weiss BG et al. [10], the 5-year estimated disease specific survival of oropharyngeal cancers after TLM was 92.8% (stage I), 85.7% (stage II), 72.5% (stage III), and 73.3% (stage IVa); the 5-year recurrence free survival was 69.1% (stage I), 49.6% (stage II), 58.8% (stage III), and 63.9% (stage Iva). Although it is difficult to compare our results with other studies because of variable study designs. It seems that the results of TORS are not inferior to those of TLM. However, TORS could conquer the limitations of TLM, such as the line-of-sight limitation when using CO2 laser beam via a narrow laryngoscope, difficult cut margin evaluation due to piecemeal resection, and poor hemostatic function for bigger vessels. The cost of robotic surgery is still high at the present time. However, if we adopted TORS for aggressively treating head neck cancers with good outcomes, we believe more economical robotic systems may undergo marketing developments and corporate competition. The cost should be reduced in the future. 

### 4.6. Study Limitations

This study had several limitations. First, the cohort size was small. Further large-scale or multi-institutional studies are necessary to confirm our results. However, our small cohort had a medium-term follow up; the evidence level is higher in our prospective study than in retrospective studies. Second, the tumors were not evenly distributed among the various anatomical sites, and most of our studied patients had hypopharyngeal cancers. However, patients with hypopharyngeal cancer were deemed to have the worst prognosis among all patients with head and neck cancer. Our results are, therefore, very encouraging. Third, the NACT was not randomized in this study. Randomization should be mandatory in future evaluations of the value of NACT before TORS. Fourth, there was still no standard guideline for us to select good candidates for transoral surgery. However, according to our experiences, surgical safety and expected functional preservation are key considerations before selecting a good candidate to receive this management. For example, the tumor border close to the carotid artery in the initial image study was a contraindication of TORS for us. If the wound without a flap covering did not heal well, postoperative massive bleeding could be a life-threatening disaster. Similarly, if hypopharyngeal cancer involved postcricoid mucosa or both arytenoids, extensive resection may cause bilateral vocal fold motion impairment and jeopardize the laryngeal functions. Although it may not be easy, continuously reporting and sharing experiences of TORS is the only way to obtain consensus in the future. 

## 5. Conclusions

TORS is an effective organ-preserving, minimally invasive surgery for cancer of the pharynx and larynx. In more than half of our patients, irradiation could be avoided, or the dosage could be reduced to about 6000 cGY during the treatment of cancer. Most patients died of neck recurrence and distant metastasis, but not primary recurrence. The medium-term 5-year overall survival rate of 75.5%, disease-specific survival rate of 86.3%, and recurrence-free survival rate of 71.1% are promising. Patients had a high organ preservation rate and good medium-term functions. NACT was shown to be a promising method for facilitating tumor resection in TORS and merits further investigation.

## Figures and Tables

**Figure 1 jcm-10-00967-f001:**
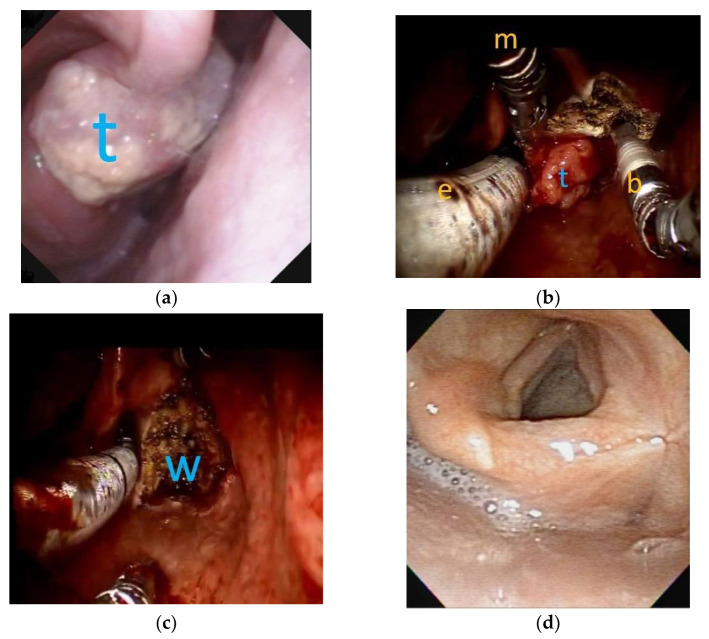
The flexible laryngoscopic view and robotic surgery view of a T2 right side arytnoepiglottic fold cancer: (**a**) before transoral robotic surgery (TORS), a clinical T2 tumor (t) was located at right-side arytenoepiglottic fold; (**b**) during TORS, under general anesthesia with oroendotracheal tube (e) ventilation, robotic Maryland dissector (m) and monopolar bovie (b) were used to resect the tumor. (**c**) Complete enbloc resection of the tumor was achieved by TORS with surgical wound (w) on the right pyriform sinus. (**d**) Three years after TORS without adjuvant radiotherapy, the pyriform sinus exhibited good appearance without tumor recurrence.

**Figure 2 jcm-10-00967-f002:**
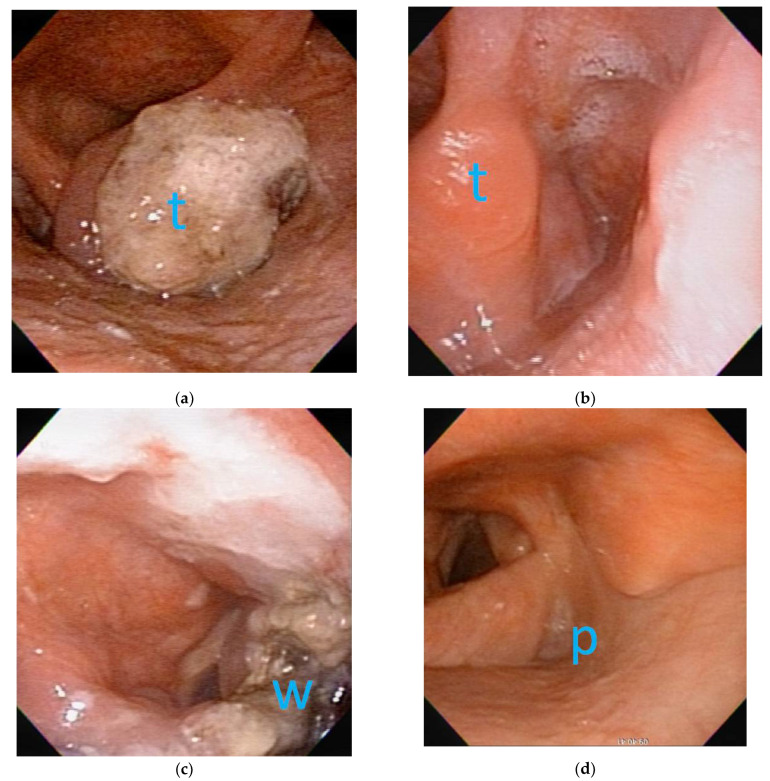
The flexible laryngoscopic images before and after neoadjuvant chemotherapy (NACT) and transoral robotic surgery (TORS):(**a**) before NACT, a clinical T3 tumor (t) was located at right-side arytenoepiglottic fold with impaired right vocal fold motion; (**b**) after NACT, complete remission of tumor (t) was achieved with no obvious tumor seen on right-side arytenoepiglottic fold. (**c**) Three days after TORS, the wound coated with fibrin showed the excision extent in accordance with original tumor border. (**d**) Three years after TORS without adjuvant radiotherapy, the pyriform sinus (p) exhibited good appearance without tumor recurrence.

**Figure 3 jcm-10-00967-f003:**
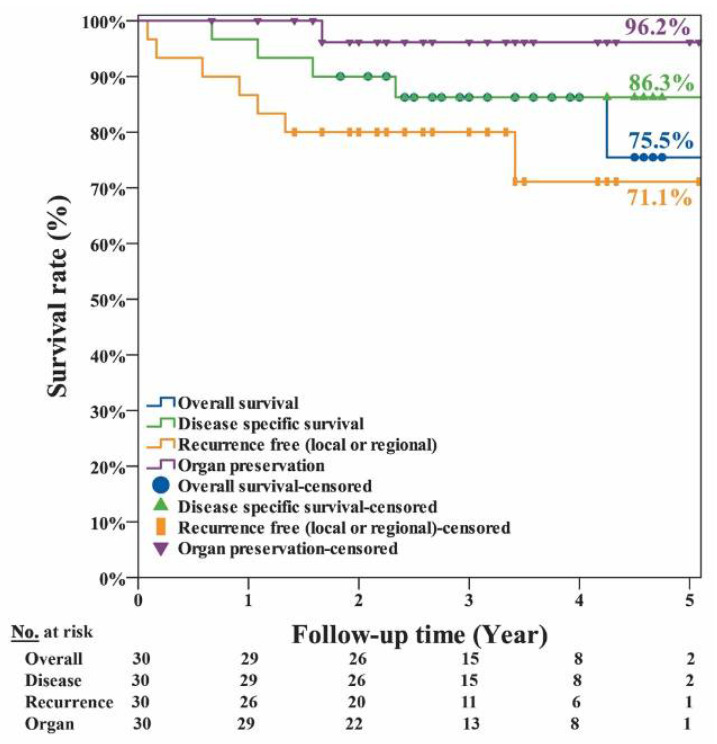
The estimated 5-year overall survival rate was 75.5%; the disease-specific survival rate was 86.3%; the recurrence-free survival rate was 71.1%; and organ preservation rate was 96.2% for the 30 patient cohort of pharyngeal and laryngeal cancers.

**Figure 4 jcm-10-00967-f004:**
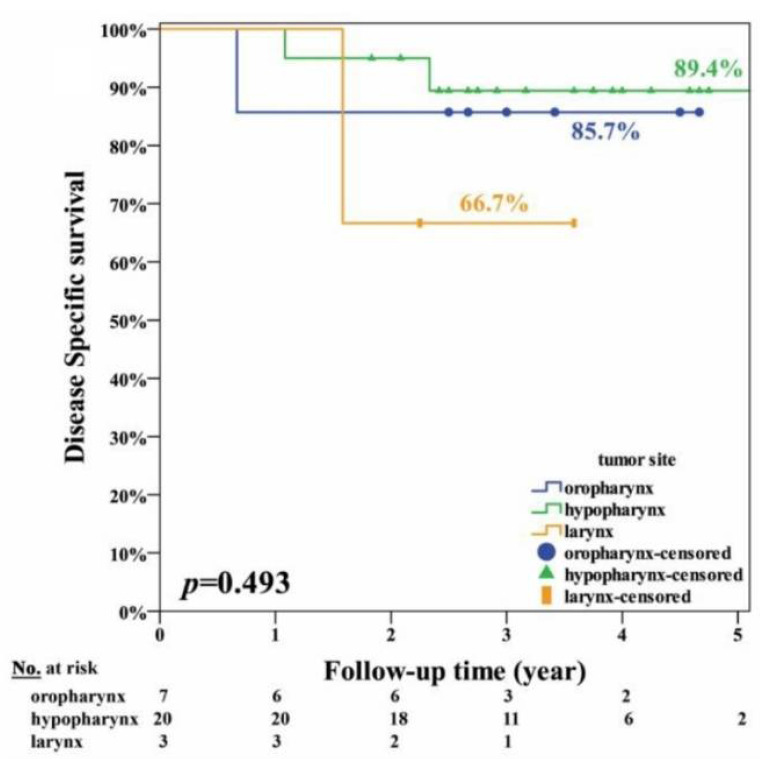
The disease-specific survival curves of 30 patients with different tumor sites including oropharynx (*n* = 7), hypopharynx (*n* = 20), and larynx (*n* = 3).

**Table 1 jcm-10-00967-t001:** The clinical factors and survival of 30 patients with pharyngeal and laryngeal cancers.

	Total (*n* = 30)		Overall Survival				*p* Value
			Survival (*n* = 25)		Death (*n* = 5)		
	*n*	%	*n*	%	*n*	%	
Age (mean ± SD)	55.90	±10.14	56.60	±10.43	52.40	±8.62	0.407
sex ^f^							1.000
Female	2	(6.67%)	2	(8.00%)	0	(0%)	
Male	28	(93.33%)	23	(92.00%)	5	(100%)	
Tumor site							0.716
oropharynx	7	(23.33%)	6	(24.00%)	1	(20.00%)	
hypopharynx	20	(66.67%)	17	(68.00%)	3	(60.00%)	
larynx	3	(10.00%)	2	(8.00%)	1	(20.00%)	
Clinical T stage							0.373
1	13	(43.33%)	12	(48.00%)	1	(20.00%)	
2	11	(36.67%)	9	(36.00%)	2	(40.00%)	
3	6	(20.00%)	4	(16.00%)	2	(40.00%)	
Clinical N stage							0.940
0	13	(43.33%)	11	(44.00%)	2	(40.00%)	
1	7	(23.33%)	6	(24.00%)	1	(20.00%)	
2	10	(33.33%)	8	(32.00%)	2	(40.00%)	
Clinical overall stage							0.981
1	6	(20.00%)	5	(20.00%)	1	(20.00%)	
2	6	(20.00%)	5	(20.00%)	1	(20.00%)	
3	8	(26.67%)	7	(28.00%)	1	(20.00%)	
4	10	(33.33%)	8	(32.00%)	2	(40.00%)	
Neoadjuvant chemotherapy ^f^							0.327
No	19	(63.33%)	17	(68.00%)	2	(40.00%)	
Yes	11	(36.67%)	8	(32.00%)	3	(60.00%)	
Pathologic T stage							0.151
0	6	(20.00%)	5	(20.00%)	1	(20.00%)	
1	17	(56.67%)	15	(60.00%)	2	(40.00%)	
2	6	(20.00%)	5	(20.00%)	1	(20.00%)	
3	1	(3.33%)	0	(0%)	1	(20.00%)	
Pathologic N stage							0.373
0	15	(50.00%)	12	(48.00%)	3	(60.00%)	
1	5	(16.67%)	5	(20.00%)	0	(0%)	
2	6	(20.00%)	4	(16.00%)	2	(40.00%)	
X	4	(13.33%)	4	(16.00%)	0	(0%)	
Pathologic overall stage							0.832
1	9	(30.00%)	7	(28.00%)	2	(40.00%)	
2	4	(13.33%)	3	(12.00%)	1	(20.00%)	
3	5	(16.67%)	5	(20.00%)	0	(0%)	
4	5	(16.67%)	4	(16.00%)	1	(20.00%)	
X	7	(23.33%)	6	(24.00%)	1	(20.00%)	
Pathologic cut margin ^f^							0.565
free	23	(76.67%)	20	(80.00%)	3	(60.00%)	
With concern	7	(23.33%)	5	(20.00%)	2	(40.00%)	
Adjuvant radiation ± chemotherapy ^f^							
No	18	(60.00%)	14	(56.00%)	4	(80.00%)	
Yes	12	(40.00%)	11	(44.00%)	1	(20.00%)	
Local recurrence ^f^							0.119
No	26	(86.67%)	23	(92.00%)	3	(60.00%)	
Yes	4	(13.33%)	2	(8.00%)	2	(40.00%)	
Regional recurrence^f^							0.001 **
No	25	(83.33%)	24	(96.00%)	1	(20.00%)	
Yes	5	(16.67%)	1	(4.00%)	4	(80.00%)	
Distant metastasis ^f^							<0.001 **
No	26	(86.67%)	25	(100%)	1	(20.00%)	
Yes	4	(13.33%)	0	(0%)	4	(80.00%)	
Organ preservation ^f^							0.167
Yes	29	(96.67%)	25	(100%)	4	(80.00%)	
No	1	(3.33%)	0	(0%)	1	(20.00%)	
Functional Outcome Swallowing Scale ^f^				*n* = 25			
0			22	(88.00%)			
1			3	(12.00%)			
Voice Handicap Index-10							
0			20	(80.00%)			
2			2	(8.00%)			
7			1	(4.00%)			
10			1	(4.00%)			
17			1	(4.00%)			

T test. Chi-Square test. ^f^ Fisher’s Exact test. ** *p*<0.01.

**Table 2 jcm-10-00967-t002:** Clinical/pathologic T & N stages for 11 patients of hypopharyngeal cancer receiving neoadjuvant chemotherapy and their post-TORS management and follow-up data.

Patient No.	c Stage	cT Stage	pT Stage	cN Stage	LN Remission	pN Stage	RT (cGY)	Local Recurrence Time	Death Time
1	II	2	1	0	-	0	-	-	51 months ^#^
3	II	2	1	0	-	0	6000	-	-
4	III	2	1	1	partial	0	-	-	13 months ^##^
10	IV	3	0	2	partial	2	6000	20 months	28 months ^###^
16	IV	3	1	2	partial	1	6000	-	-
17	III	3	0	1	partial	0	-	-	-
21	III	1	0	1	partial	0	5000	-	-
23	II	2	0	1	partial	0	-	-	-
27	III	2	1	2	partial	1	6600	-	-
29	III	3	0	1	partial	0	-	-	-
30	III	3	0	0	-	Na *	-	-	-

* na: patient rejected neck dissection due to initial clinical N0 stage. ^#^ Patient had second esophageal cancer 3 years after TORS and following mediastinum metastasis occurred after surgery for esophageal cancer. ^##^ Patient died from neck recurrence with liver and lung metastasis. ^###^ Patient had contralateral neck recurrence at 11 months and local recurrence at 20 months. TORS, Transoral robotic surgery; LN, lymph node; RT, radiotherapy; cGY, centigray.

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
