# Peer review of "Transoral Robotic Surgery for Pharyngeal and Laryngeal Cancers—A Prospective Medium-Term Study"

_jcm, 2021, doi:10.3390/jcm10050967_

Round 1
Reviewer 1 Report
This is a very interesting study and requires further investigation.
There are some areas that should be elaborated upon. In those patients with a complete response after NACT, they still undergo TORS. I note Fig 2 however, performing a resection of an area with no visible tumor may cause some to wonder if there was any cancer present at the time. Was a biopsy performed at the time or frozen section? Did any of the patient s with a complete response at the primary have neck metastases that also have a complete response within the neck?
I am aware that the numbers of patients in this group are small ie 11 with 6 complete responses but teasing out this group in detail is important. In most other centers, this group would have received radiotherapy or possibly CRT without any operation.
Under pathology and 'Cut Margin' what does 'with concern' mean? Did the frozen section reveal positive margins at the time of surgery requiring further resection or was the final pathology positive? This requires further explanation and will give the reader an understanding with post operative adjuvant treatment was needed.
Reviewer 2 Report
This is a prospective study on the outcomes of robotic surgery performed via a transoral route for upper airway digestive cancers in three sites over a three-and-a-half year period at one institution. The study also evaluates the use of neoadjuvant chemotherapy and adjuvant radiation. A total of 30 patients were included. It was found that neoadjuvant therapy with cisplatin was successful in some tumor reduction in all 11 patients it was used in and helped to achieve negative margins. Also, the need for adjuvant radiation was reduced – given in 40 % of patients. This is a useful analysis of this relatively new approach. I have several specific questions and comments:
- The methods mention that surveillance was done for the patients postoperatively and which investigations were used, line 109-112. However, it would be helpful to state how often these follow-up investigations were done.
- The authors state that main purpose of the study was to establish long-term survival, in line 124 and in the title and abstract---however, the mean follow-up was 3 years and 3 months. How is long-term being defined, considering almost half the patients had early stage cancer with expected 5-year survival of 85-90%? Would it be more appropriate to call this medium-term survival?
- Neck dissection- either unilateral or bilateral- was performed in a majority of patients. Further information on these dissections would be helpful e.g. what was the extent/levels of these dissections, how many lymph nodes were excised and what percentage were positive? Although the information on N-stage is included in table 1, the specifics would be of benefit to readers who wish to understand the robotic transoral technique for such dissections.
- Five patients died during follow-up, four had recurrence at the local site, five had locoregional recurrence in the neck. Although some general data is included in the figures 3 and 4, specific timing of these recurrences would be helpful in results section 3.3.
- It would be helpful to include comparisons of the outcomes and survival data from this study to that when other surgical approaches are used.
- Given that neoadjuvant chemotherapy helped to downsize tumors to allow minimally invasive surgery, what factors/anatomical considerations should be considered when evaluating whether it is safe to proceed with the minimally invasive approach for these tumors?
